# Evaluation of Strategies to Reduce Equine Strongyle Infective Larvae on Pasture and Study of Larval Migration and Overwintering in a Nordic Climate

**DOI:** 10.3390/ani12223093

**Published:** 2022-11-10

**Authors:** Eva Osterman-Lind, Ylva Hedberg Alm, Hillevi Hassler, Hanna Wilderoth, Helena Thorolfson, Eva Tydén

**Affiliations:** 1Department of Microbiology, Section for Parasitology, National Veterinary Institute (SVA), SE-751 89 Uppsala, Sweden; 2Department of Biomedical Science and Veterinary Public Health, Parasitology Unit, Swedish University of Agricultural Sciences, SE-750 07 Uppsala, Sweden; 3Horse Clinic, University Animal Hospital, Swedish University of Agricultural Sciences, SE-750 07 Uppsala, Sweden; 4Mälaren Equine Clinic AB, Hälgesta 1, SE-193 91 Sigtuna, Sweden; 5Department of Clinical Sciences, Swedish University of Agricultural Sciences, SE-750 07 Uppsala, Sweden; 6Realgymnasiet, Trekantsvägen 3, SE-117 43 Stockholm, Sweden

**Keywords:** Cyathostominae, *Strongylus vulgaris*, strongyle larvae, pasture management, faecal removal, harrowing, larval migration, overwintering

## Abstract

**Simple Summary:**

All grazing horses are exposed to parasites that when ingested have the potential to cause disease. Drugs designed to kill parasites in horses have been used extensively since the 1960s, but this intensive use has led to the development of drug resistance, emphasising the need for more sustainable methods to lessen parasite burdens. The efficacy and suitability of pasture-management methods aiming to reduce the level of parasitic larvae in the herbage are, however, dependent on the prevailing weather conditions. The aim of the present study was therefore to generate data on the effect of regular faecal removal and harrowing of the pasture on reducing the number of parasites in herbage in a Nordic climate. Furthermore, the ability of parasite larvae to migrate from faeces to the herbage and to survive the winter season in Sweden’s cold temperate climate was investigated. Twice-weekly faecal removal significantly reduced the number of larvae on the pasture, whereas harrowing on a single occasion in the summer under dry weather conditions did not. Parasite larvae migrated as far as 150 cm from faecal pats and were able to survive one winter season, with two years of rest from grazing horses required to achieve parasite-free pasture.

**Abstract:**

Horses, as grazing animals, are inadvertently exposed to intestinal parasites that, if not controlled, may cause disease. However, the indiscriminate use of anthelmintic drugs has led to drug resistance, highlighting the need for pasture-management practices to reduce the level of parasitic exposure and lessen reliance on drugs. The efficacy of such methods depends both on the epidemiology of the parasites and the prevailing weather conditions. The aim of the study was to investigate the effect of faecal removal and harrowing on reducing the number of parasite larvae in herbage. Moreover, the migratory and survival ability of strongyle larvae in a Nordic climate was studied. Faeces from horses naturally infected with strongyle nematodes were used to contaminate pastures and grass samples were collected to harvest larvae. Twice-weekly faecal removal significantly reduced larval yields, whereas harrowing on a single occasion under dry weather conditions in the summer did not. Strongyle larvae were able to migrate 150 cm from the faecal pats, but most larvae were found within 50 cm. Both Cyathostominae and *S. vulgaris* survived the winter months with larvae harvested up to 17–18 months after faecal placement. Resting of pastures for one year greatly reduced the parasite level, but two years of rest were required for parasite-free pasture.

## 1. Introduction

Like other grazing animals, horses are exposed to intestinal parasites that have the potential to cause gastrointestinal disease [1,2]. When broad-spectrum anthelmintics were introduced on the market in the 1960s, the control of nematode parasites in horses was initially based on interval treatments, with an indiscriminate use of anthelmintics. However, although such frequent use of anthelmintics reduced the prevalence of the pathogenic parasite *Strongylus vulgaris* to low levels, it also led to the development of anthelmintic resistance in Cyathostominae and the equine roundworm *Parascaris* spp. [3]. Anthelmintic resistance is an emerging problem that has been reported for all three classes of anthelmintic drugs used in horses [4,5]. This underlines the need for a change towards more sustainable control approaches that limit or prevent the development of resistance. Therefore, targeted selective treatment, a strategy that has been advocated by parasitologists for two decades, is increasingly being implemented on Swedish equestrian premises. By only treating horses exceeding a chosen cut-off value, often 200 strongyle eggs per gram of faeces (EPG), markedly fewer anthelmintic treatments are administered, increasing the chance of slowing down the development and spread of anthelmintic resistance [6,7,8,9]. In addition, to reduce exposure to infective larvae on pasture, preventive control methods using various pasture-management methods are also recommended. Recent questionnaire surveys conducted in Ireland and Sweden have highlighted the need for the education of equine owners on optimal pasture management [10,11]. Moreover, the equine industry in Sweden has expanded and the number of horses is now estimated to be approximately 350,000, exceeding the number of dairy cows. With around 80% of all equestrian premises located in urban regions [12], grazing areas are often limited and over-stocked, allowing for the accumulation of parasite eggs.

There is a limited number of studies defining the most suitable and effective pasture-management practices for equestrian premises in a Nordic climate. The main objectives of this study were therefore to generate data on pasture-management methods aiming to reduce parasite burdens on pastures as well as examine free-living strongyle larvae epidemiology in a Nordic climate. To achieve this, the present study investigated the effect of two strategies to manage the abundance of strongyle larvae on pasture: (i) regular manual removal of faeces from the pasture and (ii) harrowing of the pasture during dry weather conditions. In addition, strongyle epidemiological characteristics were studied by exploring: (i) how far strongyle larvae migrate on the pasture and (ii) the ability of strongyle larvae to overwinter on the pasture.

## 2. Materials and Methods

### 2.1. Faecal Egg Counts and Larval Cultures

Horses naturally infected with Cyathostominae and *S. vulgaris* were identified for participation in the study. Strongyle faecal egg counts (FECs) were carried out using two different techniques. For the faecal removal study, 4 g of faecal samples were mixed with 56 mL of a saturated NaCl solution (density 1.21 g/cm^3^) and thereafter filtered through gauze prior to transferring the solution to a McMaster chamber for counting the nematode eggs [13]. For all other studies, a modified centrifugation-enhanced McMaster technique was used, where nematode eggs in faecal samples (3 g) in duplicates were flotated using 42 mL of a saturated NaCl solution (density of 1.18 g/cm^3^) [14]. Both techniques had a theoretical sensitivity of 50 EPG. In the overwintering study, larval cultures for determining the proportion of *S. vulgaris* and Cyathostominae in each faecal sample before placement on the pasture were performed on 50 g of faeces in duplicates, mixed with an equal volume of vermiculite (Weibulls, Sweden) according to Bellaw and Nielsen [15]. In brief, tap water was added to obtain a moist condition and samples were cultured at +20 °C for 12 days. Third-stage larvae were harvested after sedimentation for 24 h at +20 °C using the Baermann technique [16] and examined and identified under the microscope using morphological criteria [17].

### 2.2. Grass Sampling and Harvesting of Strongyle Larvae

All grass samples were collected from the pastures in the morning before 10 am. For each sample, a pinch of grass, approximately 0.5 cm in diameter, was cut with scissors close to the ground. Larvae were collected using the Baermann funnel method [16] and the total recoveries of nematodes were concentrated into 20 mL water. From each sample, triplicates of 1 mL subsamples were stained with Lugol’s iodine solution and the number of third stage larvae (L3) was counted. Grass samples were dried and weighed, and estimates of larvae per kilogram dry matter (DM) of grass were calculated (L3/kg DM). In the overwintering study, the number of L3/kg DM of faeces was also calculated. In addition, for the overwintering study, L3s were identified to genus level [17].

### 2.3. Study Design: Effect of Faecal Removal on Parasite Larvae on Pasture

In 2017, a parasite-free pasture (Knivsta Åby, latitude 59, longitude 17), approximately three hectares in size, was divided into two equal halves: A and B. From 18 June until 24 July (5 weeks), three horses infected with Cyathostominae, 300–750 EPG, were let to graze on the pasture, grazing on pasture A and pasture B on alternate days. During this five-week period, faeces were removed manually twice weekly from pasture A, but not from pasture B (Figure 1). Faecal samples from the horses were analysed at the start and at the end of the grazing period (week five) using a modified McMaster method as described in Section 2.1. One horse was removed after three weeks and was therefore not sampled at week five. Grass samples were collected in the morning by the same person every other week from 3 July until 23 October, resulting in a total of nine sampling occasions (Figure 1). Pinches of grass were collected as described in Section 2.2 every seventh step while walking over the pasture in a zigzag pattern in two directions [18], resulting in two subsamples of approximately 250 g grass for pastures A and B, respectively, at each sampling occasion (Figure 2). Thus, a total of 18 subsamples per pasture were collected over the duration of the study period. Larvae were harvested as described in Section 2.2.

### 2.4. Study Design: Effect of Harrowing on Parasite Larvae on Pasture

In the autumn of 2019, a parasite-free pasture (Knivsta Åby, latitude 59, longitude 17) was prepared by topping twice using a topper mower and harrowing for the removal of old grass. In 2020, the pasture was divided into two equal halves (15 × 40 m), designated pasture A and pasture B. Faeces from four horses shedding between 300–500 EPG were used to infect both pastures. Over a three-week period (5–20 May), the pastures were infected on 15 different occasions. On each occasion, faeces from the four included horses were collected in the morning and mixed thoroughly. Thereafter, faeces from the mixture were weighed and placed on the pasture in 2 kg pats, at a distance of one metre within and between rows. An equal number of faecal pats were placed on pastures A and B and, in total, each half was infected with 24 rows containing five faecal pats, amounting to 120 faecal pats in pastures A and B, respectively. The FECs of the faecal pats were estimated weekly during the first three weeks of the study period. On 23 June, pasture A was harrowed twice using a Zocon W6-4 greenkeeper, with a six-metre width and four rows of stars. The date of harrowing was based on a weather forecast of a low risk of rainfall within the coming week. Pasture B was left untreated and served as the control. Grass samples were collected from pastures A and B in the morning approximately every second week from 7 July 2020 until 7 July 2021. No samples were collected in December to March due to snow/ice on the pastures. In total, grass samples were collected on 19 occasions during the study period (Figure 3). Pinches of grass were collected as described in Section 2.2 every seventh step while walking over the pasture in a zigzag pattern in two directions, resulting in two subsamples of grass for pastures A and B, respectively, on each sampling occasion (Figure 2). Larvae were harvested as described in Section 2.2.

### 2.5. Study Design: Migration of Larvae

In 2018, a parasite-free pasture (Knivsta, latitude 59, longitude 17) was experimentally infected with faecal pats from one horse that excreted 550 EPG of Cyathostominae. Prior to infection, the pasture was newly cut to a height of 10 cm. Two faecal pats, each weighing 2 kg, were placed 4 metres apart on the pasture, on 6 June (four weeks prior to the first sampling occasion). Approximately every other week, from 2 July until 20 October, pinches of grass were collected as described in Section 2.2 at regular intervals of 15 cm along a circular perimeter at three different distances from the centre of each pat: 50 cm, 100 cm and 150 cm (Figure 4a,b). Larvae were harvested and the number of larvae per kilogram dry matter of grass was estimated for each zone as described in Section 2.2.

### 2.6. Study Design: Survival and Overwintering of Larvae

On 14 May 2020, a pasture free from parasites (Knivsta, latitude 59, longitude 17) was experimentally infected with faeces from ten horses naturally infected with Cyathostominae and *S. vulgaris*. Prior to infection, the pasture was cut to a height of 10 cm. From each horse, two faecal pats, each weighing 2 kg, were placed in a row at a distance of one metre between pats. Samples of 25 cl of faeces for estimating the number of L3 larvae from each pat were collected once per month from July 2020 until November 2020, after which collection was no longer possible due to the faeces being washed away from the pasture by rain. A volume measurement (cl) as opposed to weight (g) was used, due to varying water content in the faeces. The number of L3/kg DM faeces was then calculated. Grass samples were collected once per month from July 2020 until no larvae could be detected (April and May 2022). No samples were obtained in January–March 2021 or December 2021–March 2022 due to the ground being frozen and/or covered in snow (Figure 5a). Each month, pinches of grass were collected as described in Section 2.2 from ten evenly distributed places within a 50 cm diameter of each pat (Figure 5b). Larvae were harvested and the number of L3/kg DM grass for each month was calculated as described in Section 2.2. Additionally, L3 were identified as Cyathostominae or *S. vulgaris*.

### 2.7. Collection of Weather Data

Temperature and rainfall data for the study period were obtained from the Swedish Meteorological and Hydrological Institute (SMHI) weather station in Uppsala (Aut), latitude 59.8471, longitude 17.6320Ci, 23.453 m above sea level (https://www.smhi.se accessed on 1 April 2022). For each month of each study period, the average daily temperature was calculated. Daily rainfall data were used to calculate the total weekly, fortnightly and/or monthly precipitation for each study period.

### 2.8. Statistical Analysis

Data were entered into Excel (version 16.0) and at each time point the mean number of L3/kg DM grass or faeces was calculated. Statistical analyses were performed in GraphPad Prism 9.1.0 (GraphPad Software Inc., San Diego, CA, USA). To evaluate the effect of faecal removal and harrowing, mean L3/kg DM grass in pasture A and pasture B were compared using the non-parametric Mann–Whitney U test and a two-way ANOVA with Tukey’s multiple comparisons test. To compare the mean L3/kg DM grass between distances in the migration study and the mean L3/kg DM faeces and grass between months in the overwintering study, a two-way ANOVA with Tukey’s multiple comparison was used. Spearman’s correlation was used to determine correlations between larval yields and weather data (rainfall and temperature). The level of significance was set at *p* < 0.05.

## 3. Results

### 3.1. Effect of Faecal Removal on Strongyle Larvae on Pasture

The three horses grazing on the two halves of pasture had FECs of 300, 400 and 750 EPG, respectively, at the start of the study period. One horse (that had 750 EPG at the start of the study) was removed after three weeks. The two remaining horses both had FECs of 400 EPG at the end of the study period at week five. Faecal removal twice weekly in pasture A significantly reduced the overall amount of small strongyle L3 compared with the control pasture B (*p* < 0.05), with a mean number of L3/kg DM grass over the whole study period of 6.8 (±SD 17.2) L3/kg DM grass in pasture A and 3480.4 (±SD 4031.3) L3/kg DM grass in pasture B. Multiple comparisons further showed significant differences in larval yield in sampling weeks 11, 15 and 19 (Figure 6). In pasture A, L3 was detected on only two occasions: on 28 August (6 L3/kg DM grass) and 25 September (55 L3/kg DM grass). In pasture B, high numbers of L3 were detected from 28 August and onwards, with larval yield in this control pasture correlating strongly with the total amount of rainfall in the two weeks prior to each sampling occasion (Spearman’s correlation factor 0.90; *p* = 0.002), and the number of L3 remained high for the remainder of the study period. Larval yields and weather data are shown in Figure 6.

### 3.2. Effect of Harrowing on Strongyle Larvae on Pasture

Analyses of the faecal pats during the first three weeks of the study (5–20 May) showed egg counts of 150–600 EPG in week 1, 100–500 EPG in week 2 and 150–750 EPG in week 3. The average daily temperature from 10 days before until 10 days after harrowing varied between 13.7 °C and 25.3 °C. The actual rainfall over this time period corresponded well with the weather forecast of low precipitation, apart from on 16 June, one week prior to harrowing, when there was 24.8 mm of rain. Excluding this date, the time period from 10 days before until 10 days after harrowing rendered an average daily rainfall of below 1.2 mm and no rain was observed on the day of harrowing or from five days before until four days afterwards (Figure 7). The minimum and maximum average daily temperatures over the sampling period were −3.2 (±SD 6.1) °C in February 2021 and 20.6 (±SD 2.4) °C in June 2021 (Table 1).

Harrowing resulted in no overall significant difference in larval yield compared with that of the control field (*p* = 0.977). However, it did result in a significantly higher density of L3/kg DM grass in pasture A compared with pasture B in sampling week 18 (*p* < 0.01). In sampling week 26, a sudden rise in L3/kg DM grass was noted in the untreated pasture (B) (*p* < 0.0001), which appeared to correspond with an increase in rainfall in September, although no statistically significant correlation was found (*p* = 0.584) (Figure 7). For the other sampling weeks, no significant differences in the number of L3/kg DM grass were observed between the harrowed pasture (A) and the untreated pasture (B). The mean (±SD) number of L3/kg DM grass over the whole study period was 2646 (±4901) in the harrowed pasture (A) compared with 2880 (±5739) in the control pasture (B).

### 3.3. Migration of Strongyle Larvae

The highest density of L3/kg DM grass was noted in the 50 cm zone around the faeces with a mean number of 1590 (±SD 1731) L3/kg DM grass throughout the study period. In this zone, the highest larval yields were observed on the last four sampling occasions (weeks 12–18) (Figure 8). The density of L3 decreased with distance, with the mean number of L3 in the 100 cm zone reduced to 232 (±SD 246) L3/kg DM grass and further to 160 (±SD 154) L3/kg DM grass in the 150 cm zone, with larval yields in the 50 cm zone significantly higher than in the 100 cm and 150 cm zones (*p* < 0.05). There was, however, no significant difference in L3 density in the 100 cm zone compared with the 150 cm zone (*p* = 0.988). Weather data for the study period are shown in Figure 9. The data suggested a correlation between the monthly precipitation prior to sampling and total larval yield and larval yield in the 50 cm zone (r = 0.80), but this was not significant (*p* = 0.333). Monthly average temperature appeared to correlate negatively with total larval migration (r = −0.8), but again no significance was shown (*p* = 0.333).

### 3.4. Survival and Overwintering of Small and Large Strongyle Larvae

#### 3.4.1. Weather Data

The total monthly precipitation and mean monthly temperatures for the duration of the study period are illustrated in Figure 8. Temperatures fluctuated above and below 0 °C primarily during the months of January, March and April in both years (2021 and 2022) as well as in February 2022, with between twelve and twenty-two 24-hour freeze/thaw cycles during each of these months. In February 2021, only two freeze/thaw cycles occurred, due to the temperature remaining below 0 °C for most of the month. Freeze/thaw cycles were also observed from October to May both years, but were fewer in number, amounting to between one and six cycles per month. The month of December in both years had few freeze/thaw cycles, but differed in that temperatures remained largely above freezing in 2021, but below 0 °C in 2022, as reflected in the mean monthly temperatures (Figure 10).

#### 3.4.2. Faecal Samples

EPGs in the faecal samples deposited on the pasture on 14 May 2020 varied from 50 to 2325, with an average EPG (±SD) of 1355 (±785). A minority of the detected eggs were *S. vulgaris* eggs, with the percentage of *S. vulgaris* eggs out of the total number of detected strongyle eggs in the faeces varying from 0.22% to 6.44%, with an average (±SD) of 2.34 (±1.86)%. Cyathostominae and *S. vulgaris* L3 could be harvested from the faecal pats during the first five months of the study period, after which the faecal pats were washed away by rain. The highest amount of both Cyathostominae and *S. vulgaris* L3 in the faeces was found in October 2020, with an average of 571,878 (±SD 499,190) L3/kg DM faeces and 9780 (±SD 7693) L3/kg DM faeces, respectively. The proportion of L3 in grass as compared to L3 in faeces increased over the first five months from 3% to 37% and from 5% to 27%, for Cyathostominae and *S. vulgaris*, respectively (Figure 11a,b). This increase in proportion was not linear with time.

#### 3.4.3. Cyathostominae in Grass Samples

For Cyathostominae L3 in grass samples, larval yield increased significantly from July 2020 (*p* < 0.01) and August 2020 (*p* < 0.05) to September 2020, which was the month when the greatest average larval yield was found (9501 (±SD 8032) L3/kg DM grass), although there was a large variation between samples (Figure 12a). There was no significant reduction in larval counts from December 2020 (8175 (±SD 8398) L3/kg DM grass) until sampling could commence after snow and ice in April 2021 (5514 (±SD 5652) L3/kg DM grass) (*p* > 0.9999). A successive reduction in larval yield was observed over the spring months in 2021, and a significant reduction from the larval yields in September, October and December 2020 was found from June 2021 onwards (*p* < 0.01). The last month when L3 were detected in any of the samples was November 2021. From December 2021 to March 2022, sampling was again impeded by ice and snow. Sampling in April and May 2022 produced zero larval yields in all samples.

#### 3.4.4. *S. vulgaris* in Grass Samples

For *S. vulgaris* L3 in grass samples, L3 levels were substantially lower than those of Cyathostominae, with the greatest monthly larval yield found in December 2020 (80 (±SD 67) L3/kg DM grass) (Figure 12b). A significant reduction in larval yield from the amount of larvae found in October and December 2020 was observed in November 2021, when *S. vulgaris* larvae were only found in one sample (29 L3/kg DM grass) (*p* < 0.05). No significant reduction in larval counts was found from prior to freezing (December 2020) until sampling again commenced in April 2021 (75 (±SD 108) L3/kg DM grass) (*p* > 0.9999). When sampling commenced after the second winter, in April and May 2022, it showed zero larval yields.

## 4. Discussion

In order to limit the use of anthelmintic drugs, alternatives such as pasture-management methods must be employed to minimise horses’ exposure to intestinal parasites. The present study showed that by removing faeces twice weekly, a previously parasite-free pasture could remain almost completely free of infective larvae for a five-week grazing period and then for a further three months. This finding is in accordance with previous studies, where twice-weekly faecal removal has been shown to be highly effective in reducing the number of infective larvae on pasture, and even more effective than anthelmintic drugs [19,20]. Another study showed that egg-shedding from horses on pastures where faeces were removed either daily or 1–2 times per week was significantly reduced after 10–12 weeks [21]. Unfortunately, survey studies have revealed that regular faecal removal appears to be rarely employed in several countries [10,11,22,23,24,25,26,27]. In Sweden, the overall use of faecal removal was only 6% in 2007 [27], and although a higher percentage of horse owners in a more recent study reported that they removed faeces (46.2%), only a minority did so at least twice weekly (7.1%) [11]. Another recent Swedish survey showed that faecal removal at least once per week was much more common during the winter months (52.9%) than in the summer (11.8%) [28]. Although faecal removal may be performed during the winter months primarily for hygienic reasons, the results of these surveys nonetheless indicate the need for increased awareness that faecal removal as a tool to lower parasite infection levels is very effective if performed regularly during the summer months, when larval development and, consequently, the risk of infection are high [29,30]. Faecal removal has the added benefit of increasing the pasture’s grazing area, which could act as a positive reinforcement for reluctant horse owners that consider the method labour intensive. In addition, in temperate climates, which includes the southern part of Sweden, little larval development was shown to occur within the first week of faecal deposition on the pasture, and it is possible that weekly faecal removal in such climatic zones would suffice [31]. Further studies investigating the effect of lower frequencies of faecal removal in Nordic climates would therefore be of interest.

The present study showed that harrowing in mid-June in a cold temperate climate did not reduce the overall number of infective larvae during the remainder of the grazing season (July–September). In addition, no differences in larval counts from the un-harrowed pasture were found for the duration of the year following harrowing. Harrowing is nonetheless practised in temperate climates. One survey conducted in Scotland showed that harrowing of pastures was associated with a decreased frequency of anthelmintic treatments, which was thought to be related to the responders’ belief that harrowing had a positive effect on reducing parasite burdens on pasture [32]. In Sweden, one study showed that harrowing or topping the pasture was the most common pasture-management method used, being employed by 36% of respondents [27]. Unfortunately, scientific studies investigating the effect of harrowing on the number of infective larvae on pasture are limited. An older study showed that harrowing increased the number of nematodes in the intestine and arterial walls of foals, and also increased egg shedding [33], although a more recent study, conducted in Ohio, failed to show an association between harrowing and differences in FECs [34]. The data from the present study suggest that a single harrowing in the summer in a Nordic climate is not advisable, if the aim is to reduce the amount of infective larvae on the pasture. Since no differences in larval counts were observed over the entire year following harrowing, this method appears to be of no use even if the pasture is to be rested for a prolonged period of time, in contrast to what has previously been suggested [35]. However, in other climatic conditions, for example, in sub-tropical regions, this practice may still be of benefit [36,37]. In addition, harrowing in Nordic climates could be beneficial if performed after the grazing season, subjecting larvae to winter conditions without the protective effects of an intact faecal ball, although this has yet to be confirmed and specific studies are lacking [38,39].

In the present study, although migration of larvae was greatest within the 50 cm zone, somewhat surprisingly and in contrast to previous studies, larvae were found to migrate as far as 150 cm from the faecal pats. Langrová et al. [40] found the majority of larvae close to the faecal pats (within 10 cm) and the furthest migration was 30 cm, with less than 5% of the larvae observed at this distance. Similarly, Kuzmina et al. [41] found that the majority of larvae remained within the faecal pats. The current study demonstrated the highest larval yield in the 50 cm zone, in early September, which appeared to correlate with the high amount of rainfall in the previous month. There was no statistical significance in the correlation, however, presumably due to there being too few sampling points, but previous studies have found a significant positive correlation between the amount of rainfall and larval yield [40,41]. There seems to be an important interaction between temperature and rainfall, with higher average temperatures (17 °C) and rainfall resulting in a significantly greater herbage larval yield than lower temperatures (4 °C), with the mechanical dispersing of the faecal pat during rainfall suggested to be a key element in facilitating migration [40,42]. Recently, a study in bovines demonstrated a positive linear relationship between L3 migration and the amount of rainfall under otherwise controlled climatic conditions (temperature 20 °C), with at least 5 mm of rain required for migration from the faecal pats to occur [43]. Thus, in the present migration study, which was performed during the summer and early autumn, when average monthly temperatures during the first 14 weeks of the study period (July–September) ranged from 13.1 to 21.9 °C, rainfall would be expected to have a significant effect on larval migration.

Pasture-management practices, as well as the migratory ability of strongyle larvae, need to be considered in the context of horses’ grazing behaviour. Horses have been shown to divide pastures into different areas, grazing on so-called lawns and avoiding faecal pats, resulting in rough areas with tall un-grazed grass [44]. Since the distance from the rough areas to the lawns normally exceeds the migratory ability of strongyle larvae, even the greater migratory distances observed in the present study, this behaviour will effectively reduce the risk of parasitic infection [45]. However, at high stocking rates (approximately 0.2 hectare per horse) this grazing pattern was not observed, with horses grazing both lawns and roughs equally, accentuating the need for pasture-management practices, such as faecal removal, when grazing is limited [44]. Although harrowing in the present study did not significantly increase the number of L3 as compared with the control pasture, harrowing may nonetheless spread L3 further from the faecal pats and increase the risk of infection, even when stocking rates are adequate.

Both Cyathostominae and *S. vulgaris* L3 survived over the winter months and L3 could still be harvested 17–18 months after faecal placement, showing that resting pastures over one grazing-season and one winter season in a cold temperate climate will not render them free of strongyle larvae. There are several reports of freeze/thaw cycles exerting a negative effect on the survival of all larval stages and specific laboratory studies have shown that less than 1% of L3 survive five freeze/thaw occasions [46]. In the present study, however, a high frequency of freeze/thaw cycles during the first winter and early spring months (January, March, April) did not result in a significant reduction in larval counts. The present results are in closer agreement with another Swedish study [47], where high levels of bovine trichostrongyle larvae were shown to survive from faecal placement in June-October until April, and it has been speculated that the Swedish climate of relatively cool summers and cold winters favours larval survival [46]. Faeces appear to serve as a protective refuge for overwintering larvae [48], and for the first five months of the present study, the majority of L3 were found within the faeces. However, since rain washed the faeces away after five months of placement, any potential shielding from the faecal pat on larvae survival after the month of November could not be evaluated.

Although L3 survived the first winter season, larval yields were significantly reduced by June of the year after faecal placement, suggesting that pasture infectivity after one year of rest from grazing horses was greatly decreased, albeit not parasite-free. In addition, the viability of harvested L3 was not evaluated in the present study and since earlier studies have found that larvae that survived freezing had intestinal cells that were damaged [49], it is possible that the L3 did not retain the same degree of infectivity over time. Future tracer animal studies would be required to determine the actual risk of parasitic infection.

Faecal samples taken at the start of the overwintering study revealed that only a small percentage of strongyle eggs were *S. vulgaris* eggs and the number of *S. vulgaris* L3 in grass samples was considerably lower in each sampling month than that of Cyathostominae L3. This is in agreement with earlier studies, where Cyathostominae invariably predominated over *S. vulgaris* in larval cultures, with ranges from 0.16 to 10% of L3 being those of *S. vulgaris* in cultured faecal samples [15,50,51]. However, *S. vulgaris* larvae were still harvested from 50% of the samples in October 2021, 17 months after faecal deposition. Considering the greater pathogenic potential of *S. vulgaris* compared with Cyathostominae [52,53], this finding is of clinical importance, suggesting that pastures infected with *S. vulgaris* should be rested for a period of two grazing seasons in order to limit the risk of infection and potentially fatal non-strangulating intestinal infarction. However, this would require access to additional pasture and may not be feasible in many urban regions.

## 5. Conclusions

In conclusion, the present study highlighted the effectiveness of faecal removal from the pasture in reducing strongyle infectivity levels. This management strategy has significant potential in that it allows for a reduction in anthelmintic treatments in horses, and thus alleviates the selection pressure for anthelmintic resistance, without compromising the horses’ health. The management practice of harrowing on a single occasion during the grazing season in a Nordic climate did not reduce the number of infective larvae, despite resting the pasture.. Although the infective strongyle larval stage was able to survive one winter season, larvae numbers were significantly reduced by the next summer, suggesting that resting pastures for one year in a cold temperate climate will result in a substantially reduced infectivity level. For parasite-free pastures, however, they should be rested from grazing horses for a period of two years.

## Figures and Tables

**Figure 1 animals-12-03093-f001:**
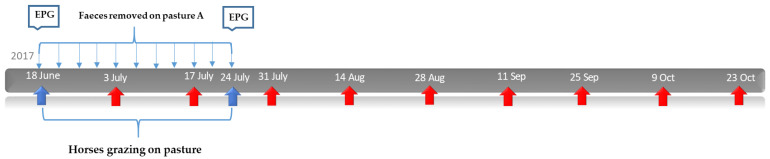
**Faecal removal study, 2017**: Timeline over the study design. Horses were allowed to graze every other day in pastures A and B, respectively, during a five week period, with number of eggs per gram (EPG) obtained at the start and end of the grazing period. Faeces were removed twice weekly from pasture A only during this five week period (narrow blue arrows). Grass samples for harvesting third stage larvae (L3) were collected every other week, starting on 3 July (red arrows).

**Figure 2 animals-12-03093-f002:**
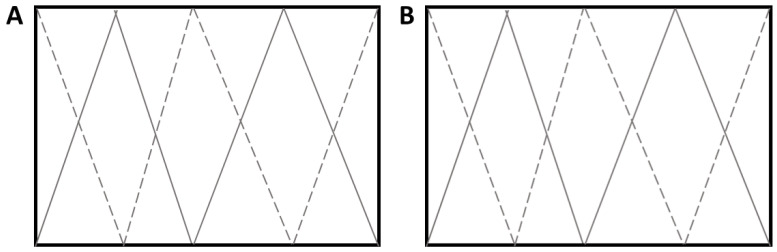
Schematic overview of the grass sampling (pasture A and pasture B) in the faecal removal and harrowing studies. (**A**) Pasture A had faeces removed/was harrowed and (**B**) pasture B served as the control. Grass was collected by walking over each pasture in a zigzag pattern in two directions (dashed vs. solid line). A pinch of grass, approximately 0.5 cm in diameter, was cut close to the ground every seventh step.

**Figure 3 animals-12-03093-f003:**
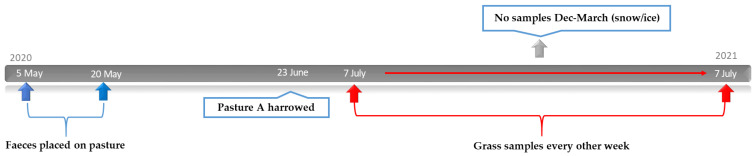
**Harrowing study, 2020–2021**: Faecal pats (in total 120 per pasture) were placed on pastures A and B on 15 occasions between 5 May and 20 May. On 23 June, pasture A was harrowed on a single occasion. From 7 July 2020 until 7 July 2021, grass samples for harvesting L3 were collected form pastures A and B approximately every second week, apart from December 2020–March 2021, due to snow and ice covering the ground.

**Figure 4 animals-12-03093-f004:**
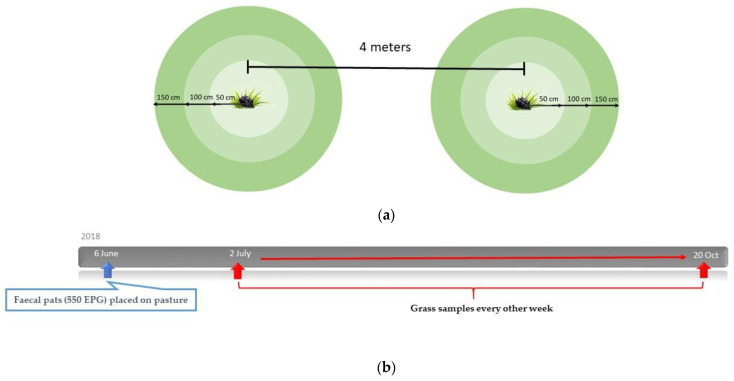
**Migration study, 2018**: (**a**) Two faecal pats, each weighing 2 kg, were placed 4 metres apart on the pasture, on 6 June (four weeks prior to the first sampling). Every other week, from the 2 July until the 20 October, pinches of grass were collected at regular intervals of 15 cm along a circular perimeter at three different distances from the centre of each pat: 50 cm, 100 cm and 150 cm. (**b**) A timeline representing an overview of the experimental set-up.

**Figure 5 animals-12-03093-f005:**
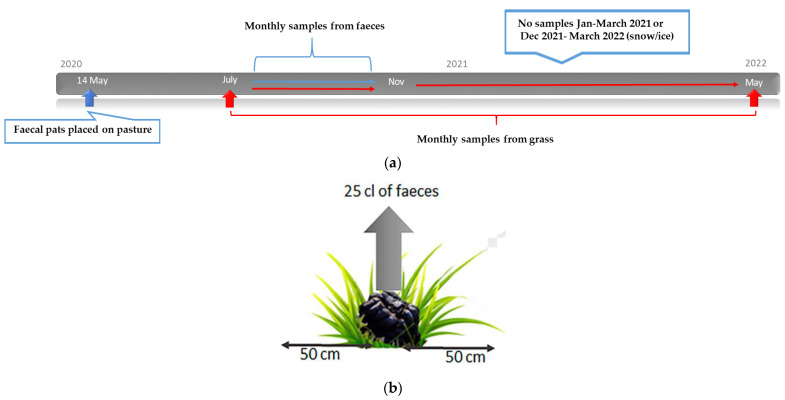
**Overwintering study, 2020–2022:** (**a**) Faecal pats (in total 20 pats) were placed on a parasite-free pasture on 14 May 2020. Faecal samples for harvesting L3 were collected every month from July–November 2020 and grass samples for harvesting L3 were collected every month from July 2020 until May 2022, apart from when the ground was covered with snow or ice. (**b**) A pinch of grass, approximately 0.5 cm in diameter, was cut from ten evenly distributed places within a 50 cm diameter zone of each pat during each month of the study period. In addition, for the first five months, faecal samples (25 cl) were obtained every month.

**Figure 6 animals-12-03093-f006:**
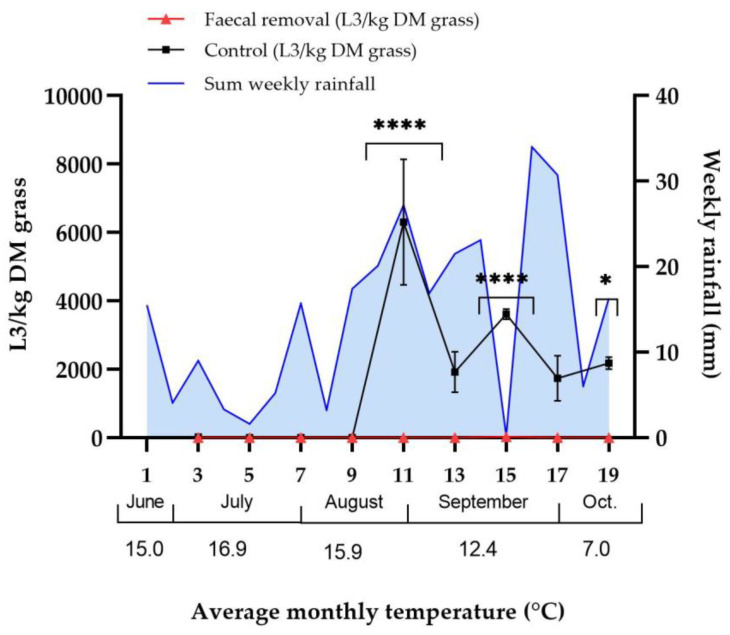
**Faecal removal study, 2017:** Larval yield (mean L3/kg dry matter of grass (DM) ±SD) in the faecal removal pasture (pasture A) and control pasture (pasture B) and weekly rainfall (mm) in each sampling week (designated 1–19 on the x-axis). The average monthly temperature is shown below the x-axis. * *p* < 0.05; **** *p* < 0.0001. Due to zero larval yield at all time points in pasture A apart from sampling points 11 and 15 (6 and 55 L3/kg DM, respectively), no error bars are shown for pasture A.

**Figure 7 animals-12-03093-f007:**
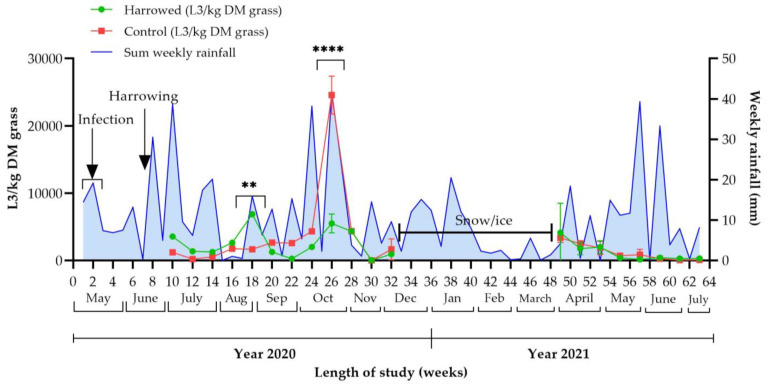
**Harrowing study, 2020–2021:** Larval yield (mean kg DM grass ±SD) in the harrowed pasture (pasture A) and control pasture (pasture B) and weekly rainfall (mm) in each sampling week (1–63). Time of infection and harrowing are indicated with arrows. No sampling was performed during weeks 33–48 due to the ground being frozen. ** *p* < 0.01; **** *p* < 0.0001. All error bars cannot be graphically shown due to very low variation between readings.

**Figure 8 animals-12-03093-f008:**
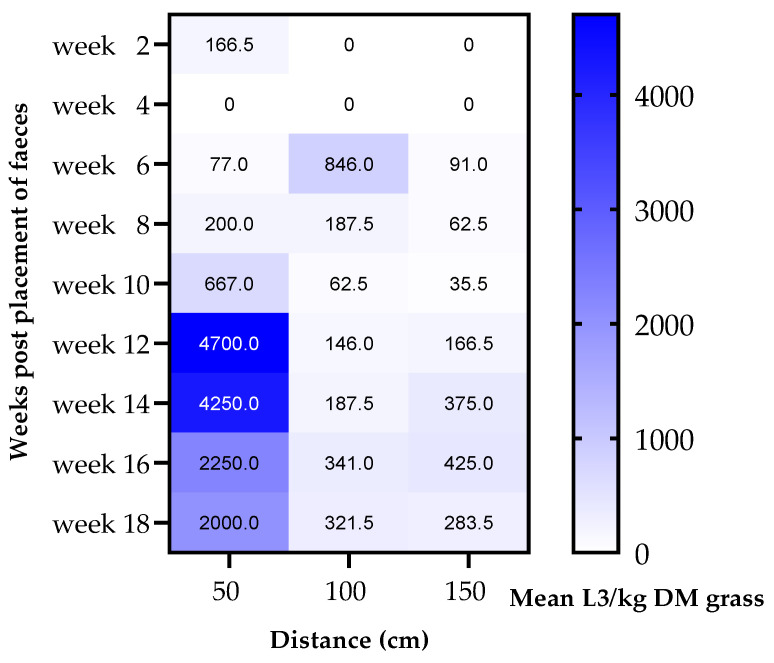
Heat map showing larval yield (mean L3/kg DM grass) at each distance zone from the faecal pats (50, 100 and 150 cm) during each sampling week of the migration study.

**Figure 9 animals-12-03093-f009:**
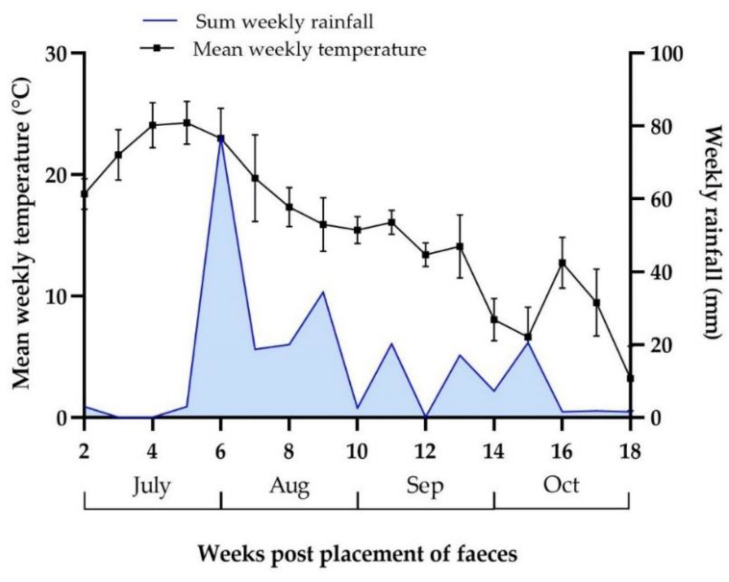
Weather data during the migration study, showing the mean (±SD) weekly temperature (°C) and the sum of weekly rainfall (mm).

**Figure 10 animals-12-03093-f010:**
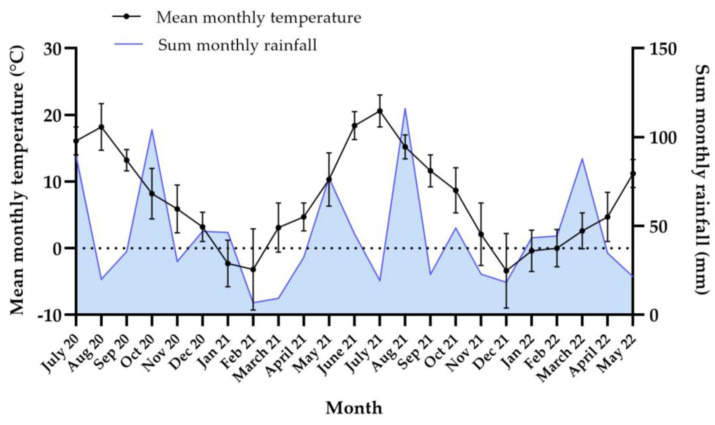
Graph showing mean (±SD) monthly temperature (°C) and total monthly rainfall (mm) during the overwintering study. The dotted gridline marks 0 °C.

**Figure 11 animals-12-03093-f011:**
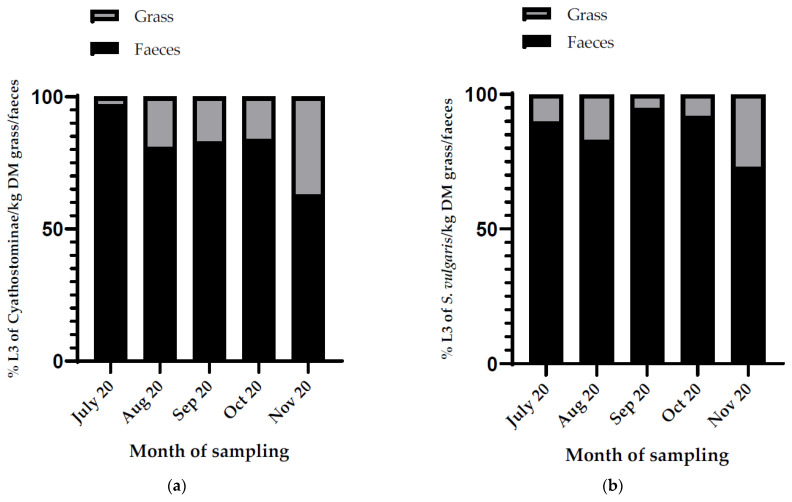
Graph showing the proportion (in %) of: (**a**) Cyathostominae larvae (L3) per kg DM faeces and grass and (**b**) *S. vulgaris* larvae (L3) per kg DM faeces and grass during the first five months of the overwintering study.

**Figure 12 animals-12-03093-f012:**
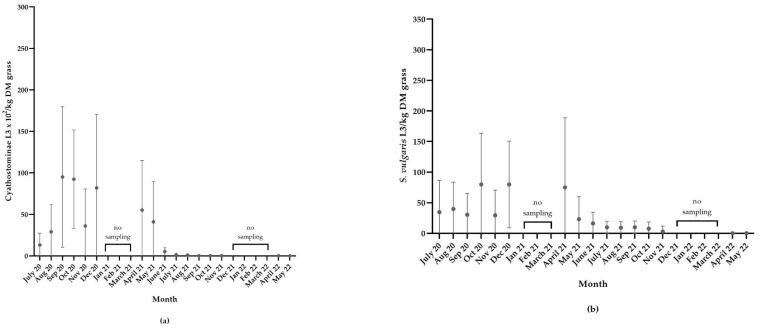
Scatter plot showing: (**a**) Cyathostominae larval yield (mean (±SD) L3 × 10^2^/kg DM grass) and (**b**) *S. vulgaris* larval yield (mean (±SD) L3/kg DM grass) during the overwintering study. Note the different scales on the y-axis for Cyathostominae (L3 × 10^2^/kg DM) and *S. vulgaris* (L3/kg DM).

**Table 1 animals-12-03093-t001:** Average (±SD) temperature during the harrowing study.

Year	Jan.	Feb.	March	April	May	June	July	Aug.	Sep.	Oct.	Nov.	Dec.
2020	---	---	---	---	9.2 (3.1)	18.4 (3.4)	16.1 (2.1)	18.2 (3.5)	13.2 (1.6)	8.2 (3.8)	5.9 (3.6)	3.2 (2.2)
2021	−2.3 (3.5)	−3.2 (6.1)	3.1 (3.7)	4.7 (2.1)	10.3 (4.0)	18.4 (2.1)	20.6 (2.4)	---	---	---	---	---

## Data Availability

Data supporting reported results can be found in the Appendix A: Original data—alternative approaches.xl. Weather data can be accessed at: https://www.smhi.se (accessed at 1 April 2022).

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
