# Peer review of "Evaluation of Strategies to Reduce Equine Strongyle Infective Larvae on Pasture and Study of Larval Migration and Overwintering in a Nordic Climate"

_animals, 2022, doi:10.3390/ani12223093_

Round 1

Reviewer 1 Report

Excellent research, brilliantly presented.

Question - line 106: (17) Russell 1948 - is this the correct citation here? Maybe one less distant in time.

line 88: please add the volume of NaCl solution. 42ml?

Author Response

Point-to point reply to reviewers’ comments

Reviewer one:

Question - line 106: (17) Russell 1948 - is this the correct citation here? Maybe one less distant in time. As suggested, the citation “Russell 1948”, has been replaced with a more recent (1986) reference: 

Thienpont, D.; Rochette, F.; Vanparijs, O.F.J. Diagnosing Helminthiasis through Coprological Examination, 2nd ed.; Janssen Research Foundation: Beerse, Belgium, 1986; 205 pp.

 - Line 88: please add the volume of NaCl solution. 42ml? This has been added to the materials and methods section.

Reviewer 2 Report

Revision Article: “Alternative approaches to control equine strongyle parasites” (animals-1978754)

The manuscript describes a series of studies that evaluated the effect of different faecal removal/disruption methods on the pasture contamination levels with infective larvae (L3) of horse strongyle nematodes, as well as studies to assess the migration of strongyle L3 from faecal pats and the overwintering of L3 on pasture in a Nordic climate.

The manuscript is very well written and present valuable information that can contribute to the development of practical recommendations to reduce the infectivity of pastures with horse strongyle nematodes. However, some issues need to be further detailed and clarified throughout the manuscript before it can be accepted for publication. Therefore, the following suggestions and comments are given for the different sections of the manuscript for the authors to prepare a new version.

Title:

 - The current title is vague and does not summarize the content of the manuscript. Please provide a more accurate title that describe the actual studies being presented in the manuscript. For example: “Evaluation of strategies to reduce equine strongyle infective larvae on pasture and study of larval migration and overwintering in a Nordic climate”

Abstract:

- Line (L) 25: Please indicate that harrowing was just performed once during summer.

 - L34: Replace “in the Nordic climate” by “in a Nordic climate” (see comment below).

Introduction:

 - L73: Replace “in the Nordic climate” by “in a Nordic climate”. This study was performed at a specific location in Sweden, and therefore does not represent “the” Nordic climate as a whole, but just one regional area. Please change this throughout the text.

Methods:

- L108: Please provide the region name in Sweden where the pasture used for this trial was located (with coordinates if possible).

- L109 -116: Please provide a new figure with a timeline of the study design (section 2.3), with the aim to allow readers to visually understand when the pasture was contaminated with faeces, when was the period of removal of faeces, when were the nine samplings performed, etc.

 -L115: Please replace “at five weeks” by “at week five”.

 - L129: Please provide the region name in Sweden where the pasture used for this trial was located (with coordinates if possible).

 - L131-145: Please provide a new figure with a timeline of the study design (section 2.4) to allow readers better understand when the contamination period with faeces occurred, the harrowing date, pasture sampling, etc.

 - L155: Please replace “four weeks prior to the first sampling occasion” by a specific date (e.g. XX May/June). The sentence “four weeks prior.…” could be kept in brackets.

 - Fig. 2. Migration study: Please place the 4 m line between the center of both faecal pats (on the top of the figure, not below the figure). Also, add the year when the study was conducted in the figure’s caption. In addition, please add a brief timeline of the study design (to describe mainly periods of pasture contamination and pasture sampling period) as a second sub-figure within Fig. 2.

 - L167: Please provide the region name in Sweden where the pasture used for this trial was located (with coordinates if possible).

 - L171: Why 25 cl and not 25 g of faces? Please clarify why “cl” are being used here.

 - Fig. 3: Please add a brief timeline of the study design (to describe mainly periods of pasture contamination and pasture sampling period) as a second sub-figure within Fig. 3.

- L196: What “values” do the authors were comparing here? Mean L3/kg DM grass between Pasture A vs Pasture B? Please describe in details what is being statistically compared.

- L197-199: What did the authors compare here using two-way ANOVA? The mean L3/kg DM grass between distances (for the migration study) and between months (for the overwintering study)? Please specify this in the text.

Results:

- Fig. 4: It seems from Fig. 4 that the Faecal removal treatment resulted in 0 L3/kg DM grass. Therefore, there is no need to statistically compare the L3 numbers with the control group, unless in the Faecal removal treatment few L3 were still identified, which should be described in the text. Also in the Figure’s caption please indicate which lines correspond to L3 numbers and which line describes rainfall. In addition, please indicate which measurement is used to present variation within data points (SD, SEM, other), and mention why some data points have variations and others do not. Please include the year when the study was conducted.

- Fig. 5: Please indicate in the Figure’s caption which lines correspond to L3 numbers and which line describes rainfall. Also please indicate which measurement is used to present variation within data points (SD, SEM, other), and mention why some data points have variations and others do not.

- Fig. 6: Please indicate in the Figure’s caption that values correspond to mean L3/kg DM grass.

- Fig. 7: Please indicate which line correspond to temperature and which to rainfall.

- Fig. 8: Please indicate which line correspond to temperature and which to rainfall. Why in Fig. 8 rainfall is now presented at the left y-axis, and not at the right y-axis as before? And why is now “precipitation” being used and not “rainfall”. Please keep consistency throughout the manuscript.

- L296-300: Please re-write this sentence, it is confusing.

- L305: Please replace “Cyathostominae grass samples” by “Cyathostominae L3 in grass samples”.

- L318: Please replace “S. vulgaris larval grass samples” by “S. vulgaris L3 in grass samples”.

- Fig. 10. Please add in both graphs a line representing the mean of all data points in each date, to show seasonal trends during the study period.

Discussion:

- L372-374: Based on the data, the authors cannot conclude that removal of faeces is better than harrowing of faeces to reduce the number of infective L3, because both methods were tested using different study designs. The removal of faeces was performed twice per week at the same time period when the pasture was being contaminated with faecal samples, thus preventing development from eggs to L3 in the pasture. In contrast, the harrowing was done only once in Summer (23 June) and > 30 days after the pasture was already contaminated with faecal samples, and thus, at harrowing there were likely infective L3 already developed (considering the average temperature at this period, L227) in the disrupted faecal pats. Therefore, the authors cannot compare both methods and cannot generally recommend that harrowing is not advisable in the Nordic climate. However, the authors can comment that based on their results it is not advisable to harrow contaminated pastures only once during the summer, especially after L3 have already developed in the faecal pats. Please also provide a critical reflection of the limitations of the study design for the harrowing experiment, and how future studies should be designed to truly compare the effect of harrowing and removal of faeces on pasture larval counts during the summer/grazing season.

- L378-381: This is a good point. Do the authors know if this has already been tested? If not, please indicate that this should be confirmed.

 - L421: Please replace “1 %” by “1%” (no space between number and %, see also L444).

Conclusions:

- L457-459: As mentioned before, the authors cannot conclude this based on their study design. Please mention the harrowing only once in summer (after allowing L3 to develop) did not reduce the number of L3 in the present study, but do not provide general recommendations.

 - L460: Please replace “June” by “next summer”. Keep in mind that the readers are based all over the world, and June is not summer everywhere. 

Author Response

Point-to point reply to reviewers’ comments

Reviewer two:

The current title is vague and does not summarize the content of the manuscript. Please provide a more accurate title that describe the actual studies being presented in the manuscript. For example: “Evaluation of strategies to reduce equine strongyle infective larvae on pasture and study of larval migration and overwintering in a Nordic climate”

The title has been changed as suggested.

Abstract:

- Line (L) 25: Please indicate that harrowing was just performed once during summer. In both the simple summary and abstract, we have clarified that harrowing was performed only on a single occasion.

  - L34: Replace “in the Nordic climate” by “in a Nordic climate” (see comment below).

Introduction:

 - L73: Replace “in the Nordic climate” by “in a Nordic climate”. This study was performed at a specific location in Sweden, and therefore does not represent “the” Nordic climate as a whole, but just one regional area. Please change this throughout the text.

 This has been done throughout the text.

 Methods:

- L108: Please provide the region name in Sweden where the pasture used for this trial was located (with coordinates if possible).

- L167: Please provide the region name in Sweden where the pasture used for this trial was located (with coordinates if possible).

- L129: Please provide the region name in Sweden where the pasture used for this trial was located (with coordinates if possible).

The region (Knivsta Åby) and coordinates have been added for the trials in each subsection. If preferred, we could instead add a sentence in the beginning of the materials and methods section, naming the location and coordinates of the trials, as they were all performed in the same location.

- L109 -116: Please provide a new figure with a timeline of the study design (section 2.3), with the aim to allow readers to visually understand when the pasture was contaminated with faeces, when was the period of removal of faeces, when were the nine samplings performed, etc.

- L131-145: Please provide a new figure with a timeline of the study design (section 2.4) to allow readers better understand when the contamination period with faeces occurred, the harrowing date, pasture sampling, etc.

In addition, please add a brief timeline of the study design (to describe mainly periods of pasture contamination and pasture sampling period) as a second sub-figure within Fig. 2.

- Fig. 3: Please add a brief timeline of the study design (to describe mainly periods of pasture contamination and pasture sampling period) as a second sub-figure within Fig. 3.

Timelines for each of the part studies have been added to the materials and methods section for clarification.

-L115: Please replace “at five weeks” by “at week five”.

Changes have been made accordingly.

  - L155: Please replace “four weeks prior to the first sampling occasion” by a specific date (e.g. XX May/June). The sentence “four weeks prior.…” could be kept in brackets.

Changes have been made accordingly.

- Fig. 2. Migration study: Please place the 4 m line between the center of both faecal pats (on the top of the figure, not below the figure). Also, add the year when the study was conducted in the figure’s caption.

The figure and caption have been changed as suggested.

- L171: Why 25 cl and not 25 g of faces? Please clarify why “cl” are being used here.

Since the faeces contained varying amounts of water, a volume measurement (cl) was used instead of a weight measure (g). Amount of L3 were then calculated per kilogram of dry matter.

 - L196: What “values” do the authors were comparing here? Mean L3/kg DM grass between Pasture A vs Pasture B? Please describe in details what is being statistically compared.

- L197-199: What did the authors compare here using two-way ANOVA? The mean L3/kg DM grass between distances (for the migration study) and between months (for the overwintering study)? Please specify this in the text.

This has been better clarified with a more detailed description of which comparisons were made in each part study, in the statistics section.

Results:

- Fig. 4: It seems from Fig. 4 that the Faecal removal treatment resulted in 0 L3/kg DM grass. Therefore, there is no need to statistically compare the L3 numbers with the control group, unless in the Faecal removal treatment few L3 were still identified, which should be described in the text.

This is already described in the text body, that L3 were recovered at two sampling points. To clarify the figure, we have now also in the figure caption added that L3 were in fact recovered during these two sampling points. Since some L3 were recovered, we performed a statistical comparison between the two pastures.

Also in the Figure’s caption please indicate which lines correspond to L3 numbers and which line describes rainfall. In addition, please indicate which measurement is used to present variation within data points (SD, SEM, other), and mention why some data points have variations and others do not. Please include the year when the study was conducted.

The figure has been clarified accordingly. The very low larvae yield in the faecal removal pasture and also during the first sampling points in the control field could not be shown graphically (and an explanation highlighting this has been added in the figure caption).

- Fig. 5: Please indicate in the Figure’s caption which lines correspond to L3 numbers and which line describes rainfall. Also please indicate which measurement is used to present variation within data points (SD, SEM, other), and mention why some data points have variations and others do not.

Please see reply to the above comment.

- Fig. 6: Please indicate in the Figure’s caption that values correspond to mean L3/kg DM grass. This has been added to the figure caption.

- Fig. 7: Please indicate which line correspond to temperature and which to rainfall.

- Fig. 8: Please indicate which line correspond to temperature and which to rainfall. Why in Fig. 8 rainfall is now presented at the left y-axis, and not at the right y-axis as before? And why is now “precipitation” being used and not “rainfall”. Please keep consistency throughout the manuscript.

This has been done and consistency is now preserved throughout the figures.

- L296-300: Please re-write this sentence, it is confusing.

The sentence has been re-written (see revised manuscript).

- L305: Please replace “Cyathostominae grass samples” by “Cyathostominae L3 in grass samples”.

- L318: Please replace “S. vulgaris larval grass samples” by “S. vulgaris L3 in grass samples”.

Changes have been made accordingly.

- Fig. 10. Please add in both graphs a line representing the mean of all data points in each date, to show seasonal trends during the study period.

 Unfortunately, the Graphpad programme would not allow this and therefore the scatter-plots showing individual values have therefore been replaced with plots of the mean values with error bars (SD) to better show the seasonal trends during the study.

Discussion:

- L372-374: Based on the data, the authors cannot conclude that removal of faeces is better than harrowing of faeces to reduce the number of infective L3, because both methods were tested using different study designs. The removal of faeces was performed twice per week at the same time period when the pasture was being contaminated with faecal samples, thus preventing development from eggs to L3 in the pasture. In contrast, the harrowing was done only once in Summer (23 June) and > 30 days after the pasture was already contaminated with faecal samples, and thus, at harrowing there were likely infective L3 already developed (considering the average temperature at this period, L227) in the disrupted faecal pats. Therefore, the authors cannot compare both methods and cannot generally recommend that harrowing is not advisable in the Nordic climate. However, the authors can comment that based on their results it is not advisable to harrow contaminated pastures only once during the summer, especially after L3 have already developed in the faecal pats. Please also provide a critical reflection of the limitations of the study design for the harrowing experiment, and how future studies should be designed to truly compare the effect of harrowing and removal of faeces on pasture larval counts during the summer/grazing season.

The aim of the present study was never to compare the two pasture management methods of faecal removal and harrowing directly and no such comparisons have been made in the manuscript. The study design was intended to mimic current pasture management practices and to objectively and separately evaluate their effect on larval abundance on pasture. The purpose of the two practices are in our view quite different, with faecal removal intending to remove L3 before they migrate out on to pasture and harrowing employed to increase the rate of death of exposed L3.

We do not quite understand the comment that L3 were allowed to develop prior to harrowing  Since L3 develop within 7 days during the summer months, harrowing would be required at intervals of less than 7 days, if to be performed prior L3 development. This would not be a feasible option for most horse owners and also would not enable harrowing to always be performed during dry weather. In addition, the objective of harrowing is to expose L3 to dry conditions. L3 cannot ingest nutrients and use up their energy quicker in a dry and warm environment and the intention of resting the pasture for a period of time after harrowing is to allow the L3 to die instead of being ingested. The current study showed that there was no significant difference in larval yield during a period of rest of greater than one year after harrowing as compared to a control field under the same weather conditions.

- L378-381: This is a good point. Do the authors know if this has already been tested? If not, please indicate that this should be confirmed.

This appears untested and we have added to the text that specific studies are lacking and this has not yet been confirmed.

 - L421: Please replace “1 %” by “1%” (no space between number and %, see also L444).

Changes have been made accordingly.

Conclusions:

- L457-459: As mentioned before, the authors cannot conclude this based on their study design. Please mention the harrowing only once in summer (after allowing L3 to develop) did not reduce the number of L3 in the present study, but do not provide general recommendations.

The sentence has been changed and recommendation excluded. Again, we have specified that our results are based on a single occasion of harrowing.

 - L460: Please replace “June” by “next summer”. Keep in mind that the readers are based all over the world, and June is not summer everywhere.

Changes have been made accordingly.